# Revisioning Fitness through a Relational Community of Practice: Conditions of Possibility for Access Intimacies and Body-Becoming Pedagogies through Art Making

**Meredith Bessey [1,*], K. Aly Bailey [2] , Kayla Besse [3], Carla Rice [1,4] , Salima Punjani [5] and Tara-Leigh F. McHugh [6]**

1 Department of Family Relations and Applied Nutrition, University of Guelph, Guelph, ON N1G 2W1, Canada
2 Department of Health, Aging & Society, McMaster University, Hamilton, ON L8S 4L8, Canada
3 Stratford Festival, Stratford, ON N5A 6V2, Canada
4 Re•Vision: The Centre for Art and Social Justice, University of Guelph, Guelph, ON N1G 2W1, Canada
5 Independent Self-Employed Artist and Curator, Montreal, QC H3N 1T7, Canada
6 Faculty of Physical Education & Recreation, University of Alberta, Edmonton, AB T6G 2R3, Canada
* Correspondence: besseym@uoguelph.ca

**Abstract:** *ReVisioning Fitness* is a research project and community of practice (CoP) working to reconceptualize "fitness" through a radical embrace of difference (e.g., trans, non-binary, queer, Black, people of colour, disabled, and/or fat, thick/thicc, curvy, plus sized), and a careful theorising of inclusion and access. Our collaborative and arts-based work mounts collective resistance against the dominant power relations that preclude bodymind differences within so-called "fitness" spaces. In this work, we build queer, crip, and thick/thicc alliances by centring relational and difference-affirming approaches to fitness, fostering a radical CoP that supports dissent to be voiced, access intimacies to form, and capacitating effects of body-becoming pedagogies to be set in motion. In this article, we consider how conditions of possibility both co-created and inherited by researchers, collaborators, and the research context itself contributed to what unfolded in our project and art making (multimedia storytelling). By a radical CoP, we mean that we mobilise a more relational and difference-affirming notion of CoP than others have described, which often has involved the reification of sameness and the stabilisation of hierarchies. Further, we call on leaders in fitness organisations to open conditions of possibility in their spaces to allow for alternative futures of fitness that centre difference.

**Keywords:** participatory research; fitness; communities of practice; disability arts; access intimacy; body-becoming

## 1. Introduction

Art and fitness in the western world may seem disparate, but they have at least one thing in common: both are known for precluding certain types of bodies—fat, raced, gendered, and impoverished—from accessing and enjoying recreation, leisure, physical activity, and cultural spaces (e.g., Chandler et al. 2018, forthcoming; Rinaldi et al. 2020). *ReVisioning Fitness* is a participatory arts-based research project, collective, and community of practice (CoP) made up of people with varied lived experience and expert knowledge, working together to reconceptualise "fitness" and movement practices with accessibility and inclusion in front of mind. This project brought together individuals from groups imagined as non-vital within the dominant culture (Bodies in Translation n.d.) to create a generative alliance of artists, activists, and thinkers invested in envisioning alternative futures for fitness. Our collaborative project undertaken across the northern part of Turtle Island (Canada) enacts collective resistance against eugenic and colonial forces that preclude difference within so-called fitness spaces. In this, we build on the existing efforts of queer, fat, racialised, and disabled leaders in the fitness sphere who have already undertaken the

hard work of developing and teaching non-oppressive fitness and movement practices. These community-committed organisers include Free to Move in Canada, Decolonizing Fitness based out of North Carolina, GOODBODYFEEL located in Ontario, Dianne Bondy Yoga in Canada, Fringe(ish) Fat Positive Yoga based in Toronto, and Black Girl Fitness operating out of Nova Scotia. We believe in the movement-making and -reinvigorating potential of non-normative arts and cultures to help create necessary change and disrupt ableist, racist, fatphobic, heteronormative, and cis-sexist structures (e.g., Chandler et al. 2018, forthcoming; Rice et al. 2021) that dominate gyms, yoga studios, and other spaces intended for physical activity, leisure, and culture sharing. As disability scholars Eales and Peers (2016) note, researchers have seldom explored the transformative potential of the disability and non-normative arts and culture movements to disrupt the inaccessible praxis that dominates contemporary fitness and physical activity spaces. We thus engage with and across theories of coalition building, CoPs, access intimacies, body-becoming pedagogies, and capacity/debility to understand the potential of working together with disability and non-normative communities to reimagine fitness.

Our *ReVisioning Fitness* collective has a working definition of fitness that we intentionally leave open. So far, our alternative understanding of fitness hinges on a radical embrace and celebration of different ways of moving, sensing, and perceiving our bodies that invite joy, pleasure, and connection to ourselves, each other, and the land. In this reimagining of traditional notions of fitness, we urge relational understandings of access that create open-ended and supportive possibilities for movement and embodiment and that refuse neoliberal capitalist approaches to inclusion.

Building from our working definition of fitness, in this article we consider how conditions of possibility, both co-created and inherited (e.g., through the ongoing interactions between bodies and worlds that co-constitute both) by researchers and the research context itself have contributed to our revisioning of fitness and of what is typically understood as a CoP through the lens of access intimacy (Mingus 2011). Mia Mingus (2011) understands access intimacy conceptually as "that elusive, hard to describe feeling when someone else 'gets' your access needs" (para. 4). She describes access intimacy not as acting charitably or dutifully toward disabled "others" but rather as engaging relationally across nondominant differences through building and deepening interdependent connections: "Sometimes access intimacy doesn't even mean that everything is 100% accessible... Sometimes it is someone just sitting and holding your hand while you both stare back at an inaccessible world" (para. 9). Inspired by the possibilities of access intimacies, we mobilise a more relational conceptualisation of a CoP than what has been previously described (e.g., Cox 2005). Formulated in the early 1990s to explain processes of co-learning in group contexts (Lave and Wenger 1991), CoPs have since been taken up in management research and used as a tool to foster workplace innovation (e.g., Roberts 2006). Wenger (1998) argues that three key characteristics define a CoP: instructors' and learners' mutual engagement, a joint enterprise, and a shared repertoire, all of which can contribute to sameness or homogeneity in such communities. Being engaged in a CoP often involves absorbing the habits and values of the group, which may serve to stabilise hierarchical ideas of expertise and hegemonic power relations (e.g., Cox 2005). While a few researchers have reported on CoPs that subvert norms and centre difference, conflict, and challenge (e.g., Limatius 2019), many studies have found that praxis communities tend to stabilise power and aim for consensus and uniformity, and thus unintentionally reproduce dominant ideologies and normative standards of being and doing (e.g., Contu and Willmott 2003; Curnow 2013). This has led to critiques of CoPs as lacking attention to power and social marginalisation.

Based on their tendencies towards the normative, we believe both fitness and CoPs are ripe for reimagining. We have begun to reconceptualise the idea of CoP through our collective work and our engagement with the idea of *ReVisioning Fitness* as a generative coalition. We argue that enacting a relational orientation to collaboratively politicized praxis serves as fertile ground for power-attuned CoPs and access intimacies to form in difference-affirming spaces. We see this work as an example of micro-activist affordances (Dokumaci

2019), the "ongoing and (often) ephemeral acts of world-building" that transform the daily lives of disabled, queer, fat, thicc, gendered, and racialised people in ways "that the built social and material world fails to readily provide" (p. 493). Our work centres bodies of difference in order to (re)build the fitness world via micro-activist affordances in queer, crip, and thicc ways. To inform our use of this reclaimed terminology, we draw on crip and queer theories and fat studies: similar to the liberatory reclamation of the word queer, the term crip[1] recuperates cripple, calling into question "compulsory able-bodiedness" (McRuer 2016, p. 301) with its re-inscription of a singular (able-bodied, white, hetero, male) mind-body norm (Garland-Thomson 2017; Kafer 2013; Piepzna-Samarasinha 2021; Rice et al. 2018; Viscardis et al. 2019). We also thicken fitness by actively resisting the normativity that has crept into fat studies by drawing on intersectionality (e.g., Friedman et al. 2020). Thus, we take up the alternative spelling "thicc" in this article to resist the cooptation of the term and to gesture to its roots in celebrating Black bodies that were made invisibilised, hypersexualised, or stereotyped via the forces of sexism, fatphobia, and racism (Cooper 2021; Griffin et al. 2022).

In what follows, we describe and theorise our ways of working together as a CoP with one caveat: we do not consider the iterative approaches that we introduce as a playbook for how this sort of research *should* be done. Rather, we highlight forces acting on and within the research apparatus that have allowed for new insights to surface within the time-space of our project, and we also develop a case for relational and difference-affirming CoPs wherein access intimacies and body-becoming pedagogies *may* thrive in capacitating ways. In our writing and theorising, we draw on content from our group meetings, individual interviews, and multimedia creations (digital stories) and provide author reflections on the research process.

## 2. Framework and Conditions of Possibility

In what follows, we describe the elements or forces that have acted on us, constituted the project from the fore, and opened (and likely foreclosed) possibilities for what could be created. We call these elements conditions of possibility, which include who we are as a team, our engagement with disability art making, our work of building a community of care, our collective reimagining of CoPs in the online space, and our expansive approach to coalition building, including welcoming dissent.

### 2.1. Who We Are as a Team

The *ReVisioning Fitness* team began as five white mostly cis women co-investigators, Drs. Aly Bailey, Carla Rice, Evadne Kelly, and Tara-Leigh McHugh, and research assistant Meredith Bessey, working within the disciplines of kinesiology, dance studies, feminist studies, and critical dietetics, and straddling various normative and non-normative embodiments, with some of us identifying as queer, femme, fat, disabled, mad, and/or neurodivergent, and as having (or having had) eating/dieting/exercise/general body-related distress. To actively resist fitness imperatives to fit into a compulsory heteronormative, ableist, fat-hating world, the original research team recruited six additional participant co-researchers from our personal and professional networks, seeley quest, Skylar Sookpaiboon, Kayla Besse, Bongi Dube, Paul Tshuma, and Salima Punjani, who brought expert knowledge (e.g., community activism, disability arts, accessibility consulting, and academia) and lived/living and embodied experiences (e.g., trans, non-binary, queer, Black, people of colour, disabled, and/or fat, thicc, curvy, plus sized) to the project.

We intentionally blur the line between so-called "academics" and "non-academics" to challenge and flatten hierarchies of power in the academy, which is also why we use everyone's first name throughout this paper. We recognise that participatory research can, at times, be problematic, inequitable, and extractive (e.g., Williams et al. 2020). Acknowledging this history, the grant principal investigator (PI, Aly) and co-investigators (Carla, Evadne, Tara) decided to pay co-researchers an hourly rate (aligned with our host institution's graduate-level research assistant pay scale) for their participation in all project

activities and for most of the writing that took place afterwards. Co-researchers also describe receiving non-monetary benefits, including camaraderie, new friendships (for some), opportunities for social connection during an isolating time, and a chance to reframe fitness in their/our lives. Regardless of these strategies, we recognize how systemic challenges always constrain efforts to mitigate hierarchies, which for us included bureaucratic hurdles and delays in paying participant co-researchers, harsh academic deadlines, time constraints when publishing, and more (see Williams et al. 2020 for more examples). Since this project intentionally intervenes at the vital edges of normative culture and outside existing fitness systems, we realize that structural-level impact is unlikely at this stage, but this is our intended goal for the future. Additionally, we orient to this CoP as fluid and ever-evolving, whereby we each resonate with or connect to this research in different ways at different times and where the connections in the group ebb and flow. Thus, the benefits and challenges of this project are likely not felt equally across the group or across time.

Beginning in early 2021, we met biweekly as a team in a curated online space (designed to meet and surpass the highest standards of accessibility) through Zoom, a total of eight times. These meetings involved sharing our own (positive and negative) experiences in fitness spaces, learning about the eugenic history of fitness, and discussing our ideas for how fitness's future could shift in difference-affirming ways. In her role as PI, Aly conducted one-on-one interviews with each participant co-researcher to gather individual perspectives on the topics. We also met online as a group in late summer 2021 for a co-analysis meeting. Our group continues to communicate regularly about ongoing writing projects and other knowledge-sharing opportunities (e.g., film screenings with fitness stakeholders). This praxis community had three main aims: to share and analyze our experiences of fitness; to question and subvert dominant meanings of fitness; and to imagine alternative futures for fitness through engagement with disability arts via a multimedia storytelling workshop.

### 2.2. Disability Art Making

In addition to the elements of our project outlined above, we engaged in a four-week virtual multimedia storytelling workshop in June 2021, supported by the Re•Vision Centre for Art and Social Justice at the University of Guelph (Evans et al. 2022). In this workshop, all six participant co-researchers and three of five co-investigators created multimedia stories—or short videos paired with audio recordings of first-person narratives, images, and other soundscapes. These digital stories can be a powerful way to speak back to dominant discourses and norms, challenge the stories that are told *about* people with bodymind differences, and be a powerful pedagogical tool in helping to shift cultural narratives (LaMarre and Rice 2016; Rice et al. 2015). We intend to use the stories in film screenings with various stakeholders to help disrupt conventional ideas of fitness. Story-making included an in-depth framing of the themes and issues that brought us as storytellers together (e.g., questioning fitness), and each storyteller also had support to develop whatever story was most important for them to tell in the moment. The workshop included a story circle to share initial ideas; writing exercises to develop scripts; tutorials about the use of online editing software; and access to technical, writing, and conceptual support throughout the process. To conclude the workshop, we voluntarily shared our stories in a screening, and everyone left the workshop as the creator and owner of their video. This means that the creators have autonomy in how and when their videos are published or shared (for further details on methodology, see Rice and Mündel 2018, 2019; Rice 2020). Then, in post-production, Re•Vision Centre facilitators supported creators in learning closed, open, or creative captioning[2] as a disability cultural practice.

Our centring of disability arts and culture contributed to makers going to the edges of, and making meaning from, their unique embodied needs, interests, and desires within movement practices. These conditions of possibility stemming from disability cultural practices were present in the workshop and in the broader research activities we engaged in; and notably they laid the foundation for what emerged as art making allowed us to imagine alternative futures of fitness through centring and celebrating difference.

### 2.3. Building a Community of Care

A disability cultural practice that helped establish our terms of being together entailed group consideration of the politics of "no." The capacity to say no is often stripped from disabled people via structural forces (e.g., substitute decision makers, medical ableism; e.g., Rice et al. 2022a), and in fitness spaces where non-disabled practitioners often make assumptions about what disabled bodies can and cannot (or should not) do (Saxton 2018). At an early meeting, we discussed what it meant for us to feel our no, both in our work together and in fitness spaces, where "no pain, no gain" and "don't stop" mentalities often dominate and saying no feels impossible (see Richardson/Kianewesquao and Reynolds 2014 for more about safety and saying no). We also considered how saying no may not be accessible to everyone and that power dynamics influence people's ability to refuse something—shaping when, how, and with whom we, as people who are differently embedded in uneven power, can say no. Yet, we wanted to create an open space where people could honour each other's limits and engage in capacitating and fulfilling ways rather than depleting or debilitating ones. For example, while we could financially compensate participant co-researchers for most of the work, we could not pay everyone for all writing (due to a budgeting oversight), so we instead ensured that folx could say no to this unpaid work, which some did. We draw on the concepts of capacity and debility, whereby within westernized neoliberalized democracies, certain bodies are seen as capable of being productive or of living, while others are seen as debilitated or targeted for slow death (Puar 2012). We subvert neoliberal notions of capacitating by considering how this work fulfilled and satiated us in ways beyond the market's expectations of productivity and progress, in line with Puar (2012), whereby we supported each other's "no" in resisting neoliberal academic pressures that threaten to debilitate us. We return to an exploration of what capacitated each of us near the end of this paper.

During one of our early group meetings, one participant co-researcher, Salima, named an important condition of possibility for doing good work together: "being part of a community of practice... *ReVisioning Fitness* really feeds into that, because it feels like I can still be part of a community that is centred around care." She further elaborated that "when we come together and think about accessibility, even in an online space, and care and connection is thought about throughout the whole process, that in itself is a motivating reason to be part of *ReVisioning Fitness.*" Her invitation for us to become a CoP centring on care and connection struck us as especially significant given the COVID-19 lockdown conditions we were then confronting. The phrase stuck, shaping what we have accomplished in revisioning fitness and how we now orient to our work together.

### 2.4. Online Nature of Our Work

The sudden shift to the online space in the face of the COVID-19 emergency surfaced as another powerful force acting on our research. Zoom, WeVideo, Mentimeter, and other technologies opened pathways for us to think, feel, perceive, and create together in a virtual space. We consider the pandemic coupled with these online affordances as both enabling and constraining forces in our work.

One enabling component involved intentionally creating a relaxed space that we co-constituted by enacting a disability cultural practice called "relaxed performance" (e.g., Collins et al. 2022; Jones et al. 2022). Relaxed performance has emerged in the last few decades from a growing recognition of the contributions of disability, d/Deaf, mad, ageing, and fat activist-artists to disability rights and justice movements and the need to educate the mainstream about non-normative cultural practices to improve access to art (LaMarre et al. 2021). As a cultural practice, relaxed performance aims to make art and cultural spaces more accessible through technical modifications, such as dimmed lights, warnings about and/or reductions in loud noises and pyrotechnics, the choice to move freely in and out of the space, and the creation of a "chill-out space" where people can relax, and more (LaMarre et al. 2021; Collins et al. 2022). Within *ReVisioning Fitness,* we enacted this practice by encouraging people to engage in the ways that worked for them, including

turning their cameras on or off as needed, situating themselves anywhere that made them feel comfortable (e.g., their bed), and inviting freedom to eat/drink, take breaks, and flow in and out of the online meeting space as desired. In addition, because we met online, team members could join from British Columbia, Alberta, Ontario, Quebec, and Nova Scotia, allowing us to connect across provinces in ways that would not have happened otherwise. However, we recognise that online research also required everyone to have access to a computer or phone and a stable internet connection, which may have excluded some from participating. We also acknowledge that our relaxed approach to working online (e.g., the option to turn cameras off) made perceiving dissent or someone's "no" more challenging since reading the body language or facial expressions of others was not always possible.

Another constraint of the online space surfaced in our inability to explore movement in embodied ways as we had planned to in person. For example, in the original project proposal, we planned to spend two meetings exploring body movement, which was not something we could ethically enact in the online space. While not viscerally sensing or directly engaging with our own and each other's bodies limited some activities, this ostensible constraint created a certain amount of freedom, allowing us to escape the friendly but exposing gaze of others in our explorations of fitness. For example, at our second group meeting, co-investigator Evadne described how the gaze of others interferes with her ability to feel pleasure in fitness and impedes her internal perceptual-sensory experience of movement:

> a barrier to pleasure is sometimes when I feel the gaze of others and feel that I can't fit into what their—what their idea of fitness is, and yeah, so mirrors and, you know, that's one of those things that creates that feeling of being watched as opposed to an internal experience for me...

While Evadne was not specifically speaking to the experience of online fitness, her words feel apt in demonstrating why the relaxed nature of *ReVisioning Fitness* became so important in cultivating a space where we could explore freely and subvert the gaze.

*2.5. Our Expansive Approach to Coalition Building*

Our intentional creation of a coalition across bodymind differences (Price 2015)—queer, trans, non-binary, Black, racialised, disabled, mad, neurodivergent, fat, thick/thicc, and plus sized—comprised another condition of possibility for revisioning fitness. Although some of us knew each other prior to joining the project, many of us were strangers. Despite this, and the fact that many of us have only engaged with one another online, we felt a sense of relationality and community almost immediately. In her theorising on coalition building, Alison Kafer cites Catriona Sandilands on the power of coalitional rhetoric to forge connections across seemingly incommensurate differences. Sandilands writes: "A vital moment in coalitional political rhetoric is its ability to construct connections among struggles that may be not only diverse but opposed to one another in many respects" (as quoted in Kafer 2013, p. 149). We argue that effective coalition building across embodied differences must embrace disagreement and the possibility of dissent in its goal-making. Our team came together with the common goal of refusing the damaging "fit" and "unfit" dichotomy that continues to be tethered to eugenic notions of worthy/unworthy bodies (e.g., Kelly et al. 2021). At the same time, we push against sameness, taking up a politics of coalition that embraces dissent (Lorde 1997; Samantrai 2002) rather than suppressing it.

We recognise that the benefits of coalition politics are also bound up with the difficulties of such politics (Kafer 2013). It was not always easy, but we invited disagreements, tensions, and frictions in the group because otherwise we risk not recognising our own biases, assumptions, or exclusionary habits (Kafer 2013), particularly amongst the white, cis, thin, and straight and straight-passing academics on the team. These practices included disagreements being welcomed verbally by Aly as the group meeting facilitator, embracing "no" and supporting each other in our no throughout the project (e.g., choosing not to make a video, not to share a video in publication, or not being an author on a publication), giving space for people to hold opposing views, and offering opportunities to question the

research process. For example, one participant co-researcher, Paul, contacted Aly to inquire whether participants would be compensated fairly for their time outside of the storytelling workshops when working independently on their videos. This is an example of Paul's refusal to be exploited in his art making; even though he was paid for all art making hours, exploitation of his video in the future is still a risk, which is why he declined to include his story in other academic articles we have written.

An openness to refusal sits in sharp contrast with the conventional nature of the fitness industry and traditional conceptualisations of a CoP, both of which seem to value normativity and consensus (e.g., Wenger 1998). Co-investigator Tara-Leigh, in reflexive writing for this paper, spoke to the uniqueness of *ReVisioning Fitness* and the power that came from learning together through working with and across our differences:

> *The sharing and co-learning that was facilitated among this CoP stood out to me as a particularly powerful tool for working to address the problematic nature of fitness. The ideas generated from imparting our diverse experiences were instrumental in creating new knowledge. This has led me to wonder whether consideration of the uniqueness of each person's experiences is what makes a CoP particularly powerful with respect to solving problems and facilitating best practices.*

Rather than thinking about difference as a force that undermines our CoP, we consider it generative and energising, and our work as thriving amid ambiguity and contradiction. In her reflexive writing for this paper, Meredith further elaborates on what the project space felt like for her:

> *When I think of a community of practice or care, I think about people coming together around a common goal or objective, and caring for each other through the process, even when things are sticky or challenging or hard. ReVisioning Fitness has felt like such a community, even from our very first meetings, as a place where we can imagine things together, and where we can come together to resist dominant norms and ideas about fitness, even when we might have different ideas about how exactly to make that happen. It has felt like a place of vulnerability, where I can share about myself and my experiences and feel supported and held by the people in the group.*

When friction arose in our work, permission to refuse and contest created the possibility for deeper, more generative understandings of the concepts we tried to unpack. For example, in one of our team meetings, we discussed the concept of "inclusion" in fitness and created a word-cloud of terms we associated with the phrase. Most of the words proposed were relatively positive, such as "belonging", "expansion", "exciting", "acceptance", "happy", and more. In contrast, Salima contributed words such as "inauthenticity" and "fraught." Later, in a one-on-one interview with Aly, Salima elaborated on her word choices (and what she described as a "grumpy moment" during our meeting), explaining that inclusion is painful and violent for her since fitness spaces often assume what people need without meaningfully engaging with them about their access needs. In her experience, the word inclusion is rarely used in a genuine way but rather is mobilized as a marketing ploy:

> *...with the inclusion part, in terms of it being weaponized, I think that something I'm definitely hyper-aware of is that social responsibility checkmark [version of] inclusion...pretending you care and having the scripts and knowing the right things to say, but it's really just surface level, and for me, I'd rather you not do that at all, then to pretend to seem like a good person or seem like a good company... unless you're willing to do the structural changes to actually [alter] that.*

Salima alludes to neoliberal inclusionism, describing it as a marketing strategy of commercial enterprises that asks/expects bodies of difference to assimilate to the norm through so-called "inclusion" tactics that merely expand tolerance rather than meaningfully building difference-affirming spaces (Bailey et al. 2023; Jones et al. 2022; Mitchell and Snyder 2015). In her interview, she also highlighted the entangled nature of inclusion and racial and disability justice, where hiring "folx that live at various intersections that might have a better way of connecting to people" can contribute to establishing more meaningfully

inclusive spaces. Salima goes on to note that as someone with a disability, she needs a fitness trainer who can understand both the physical and the emotional aspects of fitness, as well as her need to rest and not be constantly pushing herself. Salima's disruptive understanding of inclusion became a pivotal moment in the project, propelling us to interrogate our assumptions about inclusion and turn a critical lens on the sudden uptick of "inclusive" physical activity campaigns that surged during the pandemic. We understand contradictions around the meaning of inclusion as generative insofar as they are not easily resolvable, belie any claims of unity, sameness, or homogeneity, and urge us to push into their rubs, tensions, and clashes for clues to what may be required to create inclusive praxis. We follow disability studies' call for alliances that resist normalisation and reorient to difference as political, valuable, and integral—as necessary for imagining alternative fitness worlds where differences are acknowledged as an ever-emerging, foundational part of the flow and movement of life itself (e.g., Kafer 2013; Mingus 2011; Rice 2018; Rice et al. 2021, 2022a).

## 3. What Unfolded

Our above framework and conditions of possibility—the research team, disability art making, building a community of care, the online nature of our work, and our expansive approach to coalition building—created the container for what unfolded within *ReVisioning Fitness*. We detail the unfolding that occurred within the research space, where we imagined alternative worlds for fitness via relationality, body-becoming pedagogies, access intimacies, and attending to the capacitating effects of our approaches.

### 3.1. Imagining Alternative Worlds of Fitness

Through our work together, we (re)imagined alternative fitness worlds that centre a politics of difference. Each of us imagined the future of fitness in different ways through our multimedia stories, highlighting the heterogeneity of our work. Skylar shared their relationship with fitness through everyday activities as a non-binary trans masculine person searching for relational meanings of fitness, and Meredith centred the importance of rest and refusal. Salima's story foregrounded the potential of fitness spaces as places where people have agency, while seeley noted the importance of people's dignity within these spaces, especially in changing rooms. Paul's story described his frustrations with a lack of accessibility and accommodations in fitness spaces, while Kayla illustrated how her embodied being works with and through crip time. In her story, Bongi explored her embodied experience of discomfort in a gym as a Black, plus sized woman, while Aly's story focused on implementing everyone's ideas of reimagining fitness through stakeholder involvement, and Tara-Leigh grappled with her complex relationship with fitness, both as something that has brought her joy but that also operates as a site of ongoing and historic harm for others. These accounts highlight what we say "no" to in fitness (e.g., ableism, fatphobia, racism, imperatives of productivity, the gender binary etc.) to make way for alternative possibilities and futures that we can say "yes" to. Stories were also used alongside clips from focus group meetings to create a mini-documentary, a nine minute film showcasing a snapshot of our reimagining of fitness that various team members have already screened in both academic and public presentations.

As a group, we imagined and theorised alternative offerings to the current fitness climate. These offerings encapsulate what has unfolded in our *ReVisioning Fitness* theorizing so far: individualism → relationality, biopedagogies → body-becoming pedagogies, inclusionism → access intimacies, and debilitating → capacitating processes and approaches.

### 3.1.1. Individualism → Relationality

The current fitness climate is focused on hyper-individualism (e.g., Bailey et al. 2022), but in contrast, community and relationality are central to our revisioning of fitness. Reflecting on our work together, Meredith elaborates on the centrality of relationality in our work—of being in relation with our bodily selves, with each other, and with novel ideas

and knowledge, where we collectively think, feel, and move outside of a conventional fitness framework.

> *I wonder about how our experience can help to stretch mainstream instrumentalist notions of the purposes of a community of practice or care–not about innovating or always having to move things forward, but as about being present for oneself and each other, even across digital space and incommensurable differences. Our work also necessarily attends to power differentials in fitness spaces and within our group, pushing into challenge and dissent in ways that CoP work has not always done. I recognize, as well, that I hold power in research and fitness spaces, and I wonder about how best to use that power to hold space for difference and challenge norms in the spaces I inhabit.*

Disability arts and cultural practices served as a scaffold for our relational exploration of ideas surrounding movement and embodied agency and have given us language to activate alternative concepts of fitness and vitality through a relational lens versus an individualistic one. As a group, we have aimed to become a radical CoP—one where embodied difference and relationality rather than so-called expertise and consensus take the lead, and where normative relations are not reified, allowing knowledge to keep moving. Rather than operating from a place of "aspirational independence", we consider ourselves inherently interdependent, where the making of accessible worlds becomes a "collective human practice" that we all participate in (Valentine 2020, p. 81). Our creation of a relaxed space also likely contributed to this feeling by opening to people missing a meeting, coming late, or leaving early if needed. Members of the collective co-created a culture that welcomed each of us to engage as humans in whatever way made sense in that moment. We have found that reconceptualising CoPs along the lines described by Meredith in her above reflection and in line with the notion of a generative coalition called for within disability studies and activist movements became a necessary step in our shared work to set body-becoming pedagogies in motion.

### 3.1.2. Biopedagogies → Body-Becoming Pedagogies

CoPs that place significant value on expertise, efficiency, and productivity may become places where practitioners intentionally or unintentionally introduce and reproduce biopedagogies—expert instructions for living informed by moralities (Rice 2014; Rice et al. 2022b). Biopedagogies teach about supposedly morally sound behaviours by coding bodies and minds as good or bad, thereby urging conformity to normative standards through the entanglement of affect with expert knowledge. Our society has become totally pedagogized (Bernstein 2001; Rice et al. 2018), and as such, biopedagogies circulate across all systems, including media, family, education, and healthcare (e.g., Bailey et al. 2022; Bessey and Brady 2021; Friedman et al. 2020). Fitness-related discourses often perpetuate and reinforce biopedagogies, wherein messaging about physical activity reifies moralising ideas about worthy and unworthy bodies (e.g., Bailey et al. 2022).

As part of our relational and difference-affirming approach to CoP, we intentionally work to subvert biopedagogies and take up body-becoming pedagogies instead. Body-becoming involves the artful and improvisational possibilities of affirming embodiment and bodily differences (e.g., Rice 2015). Rather than relying on normative ideas of the human bodymind or on traditional notions of expertise, becoming pedagogies value our embodied differences and knowledge and our own felt understandings of our bodies' capacities as we explore our bodily capacities in spaces designed to proliferate them (Rice et al. 2018, 2021). Taking a disability arts and culture approach to revisioning fitness in a difference-centred and relational CoP has capacitated us by broadening the script for what fitness and movement might mean and how we can communicate alternative understandings to fitness stakeholders.

For Aly, who worked in the fitness industry for over a decade, *ReVisioning Fitness* provided space and opportunity to explore becoming (physically and otherwise) outside the punishing expectations of the fitness world and to resist pressures to practice fitness in the ways that she had undertaken previously. She reflects,

*My "fitness" journey has had to transform, from subscribing to normative impulses, to refusing heteronormative, sanist, and ableist rhetoric. As a queer woman who identifies as Mad and has episodic disability, I have had to slowly reclaim fitness for myself. I was able to do this with the coalitional work we have done together across our differences. For me, that is the power of a relational CoP that embraces difference. Prior to ReVisioning Fitness, I never had a safe place to resist the harms of fitness practices and instead had to conform to the norm the best way I could to survive. As a fitness instructor, it was my job to uphold, validate, and confirm the importance of fitness from a biomedical perspective. Not only did this stifle my own growth, but it harmed others too. A radical CoP energises you to resist what feels impossible to resist, and that is body-becoming for me. But CoP is a fluid concept, with ebbs and flows, that needs patience and nurturing and although our team felt connected almost immediately, that may not always happen.*

*ReVisioning Fitness* continues to offer a space where we can "becom[e] together", where we reflect on and tell our stories and have them listened to in new and different ways (Rice et al. 2018, p. 674). We conceptualise this becoming-together as a form of access intimacy, collective care and strength, and as a fruitful alternative to individualism and neoliberal inclusionism.

### 3.1.3. Inclusionism → Access Intimacies

In our embrace of disability politics, participant co-researcher, Kayla, brought forward the idea that access intimacies can *crip* CoPs, and counter neoliberal inclusionism that pervades fitness spaces (Mingus 2011; Mitchell and Snyder 2015). We embrace access intimacy as "the kind of eerie comfort that your disabled self feels with someone on a purely access level" (Mingus 2011, para. 4), and as an exciting way to re-think fitness practices. Access intimacy emerges from deep relationality that allows a non-normatively embodied person to relax in the presence of those who "get" their access needs (Mingus 2011). Kayla reflects on how access intimacies capacitate her physically, emotionally, and ethically:

*I don't use a medical mobility aid. Rather, I see "mobility aid" as something relational rather than a physical object. What I mean is, when I am out with a trusted friend or family member, and we understand one another's needs, they know that I will need a hand or an arm up a flight of stairs, or over an icy sidewalk. When "patterned-access intimacy" (Valentine 2020, p. 83) is present in my interpersonal relationships, I feel, to co-opt a term from the fitness world, that I have better physical and emotional endurance. Patterned access intimacy ensures that the maintenance it takes to inhabit a disabled body is more of a relay race than an individual sprint, and "helps develop an ethical orientation to the world that is relational and interdependent in nature". (Valentine 2020, p. 84)*

*ReVisioning Fitness* took place entirely digitally, so no one in the virtual room could reach out physically using touch to meet another's access needs. However, we note the body-technology interface here—the nature of our online meetings meant that the physical space of the meetings, people's homes, were already accessible to them. People had technologies at home that provided the support and rest that their bodyminds required; for Kayla, this included an electric standing desk, a robot vacuum, soft furniture, and yoga supports. In the absence of touch as a mobilising force, the politic of access intimacy may be enabled via a two-fold approach: remote meetings where we have a commitment to "'staying-with' the constant struggle of inaccessibility" (Valentine 2020, p. 84) and curating our working environments to suit the needs of our bodies and minds.

We propose that fitness stakeholders meet access needs through a process of curation. The mainstream fitness community has much to learn from disability arts. By engaging with the politics of disability arts, fitness practitioners might learn how urging homogeny and "mastery" of a physical or artistic form can become violent. For example, in Kayla's past role as Public Education Coordinator at Tangled Art + Disability, she worked with a team of disability-identified people to co-create greater accessibility in the arts world. Her work with the Tangled CoP illustrates access intimacy in action: the sharing of access knowledge

between senior and junior staff members, welcoming the disruptions of disability that lead to creative innovation, and being open to others who move or think in an unexpected way, even or especially when it is unsettling. The creation of disability artworks is necessarily collaborative; to enact access, we must draw on one another's embodied knowledge of how to make art legible to a public with a diverse range of physical, mental, and sensory needs. The work of access is iterative; a checklist model of competence or compliance will always fall short of giving *everyone* what they need (e.g., Chandler et al. 2022). This line of thinking, when applied to mainstream fitness, reminds us that there is no "right way" to move or "right way" to form habits. Instead, habits can be thought of as "physical anchors that can be used as launching points for the imagination" (Hamington 2004, p. 96, quoted in Valentine 2020, p. 91).

Disability aesthetics provides an exciting tie between access to art and fitness, in that access practices innovated in disability cultural spaces can be mobilized for reimagining access and inclusion within fitness. Disability aesthetics rejects an ableist ideal in favour of a processual relationship to "desired outcomes", end goals, and the illusion of perfection. Sean Lee's (n.d.) concept of "the crip horizon" encourages us to move "away from the typically beautiful towards 'the ugly'—towards the magnificence of our imperfections—and towards an aesthetic uniquely situated and held in disability art" (para. 6). Imperfection is antithetical to the aims of mainstream fitness, which seeks to rid us of, or at least obscure, our disabilities and other bodymind differences. The embrace of disability aesthetics played a capacitating role in our work together, as did community, rest, and support, as we explore in the next section.

3.1.4. Debilitating → Capacitating

As noted above, capable bodies comprise bodies coded as productive and capable of life (Puar 2012). Within our neoliberal political economy, disabled and other non-normative bodies become debilitated or under resourced via economic and social processes that simultaneously capacitate or enable only those bodies that reproduce the hegemonic order (white, non-disabled, straight, etc.). Puar (2012) argues that queer bodies and "other bodies heretofore construed as excessive/erroneous" can also be constructed as "on the side of capacity", enabling queerness to operate "as a machine of regenerative productivity" (p. 153) when certain people are absorbed into the body politic, recuperated as good citizens, and can thus generate wealth for the nation. However, we wonder what it might mean to think about capacity differently, outside of the constraints of neoliberal ideas of productivity. The CoP that we created welcomed in difference in a regenerative way in a move towards capacitating bodies in our revisioned approach to body-becoming fitness. Creating these sorts of supportive community spaces may be one way of enacting access intimacy within the context of fitness moving forward. Meredith reflects on what capacitates her,

> *Something that capacitates me in movement practices is community and support, which has felt lacking at many points throughout the pandemic (in part due to the requirement to shift to online fitness classes, and in part due to my former yoga studio closing and feeling severed from many members of that community). ReVisioning Fitness has often helped to fill that gap, even though we were meeting online. I think back to our conversations often as I move through my yoga practice or go for a walk or spin my legs on a bike. I often "allow" myself to move in whatever ways feel intuitive for me in that moment and encourage that for the students in my yoga classes, rather than thinking about what is "right." Being connected to a group of like-minded people who have challenged my notions of fitness continues to capacitate me in my day-to-day movement practices.*

This experience of feeling and being capacitated within *ReVisioning Fitness* was in many ways oppositional to the experience Meredith portrays in her multimedia story about colonized yoga, where she did not feel capacitated to move within her own bodily potential and did not feel empowered to say no or to disengage from movement in order to rest. Respecting no in a fitness space might mean taking a rest pose despite what the teacher is instructing, which can be challenging in a group dynamic and is likely amplified

if one already feels othered in that space. Access intimacy in the fitness context might mean creating spaces that welcome deviations from the expectation or norm, including all expressions of movement or rest at any given time. This openness might help to create an affective atmosphere wherein the collective addresses access needs explicitly and processually, taking the burden off non-normatively embodied people to have to advocate for accommodations or inclusion (Mingus 2011). Rather than having someone feel humiliated or like a burden, access intimacy can leave someone feeling free, light, and cared for (Mingus 2011).

A tangible example of enacting the capacitating effects of access intimacy that we imagined as a group was the idea of a fitness doula, or fitness facilitator (a gender-neutral alternative). As another co-investigator, Carla, explained in one of our meetings together, a fitness doula is:

> . . .somebody [who could] help me learn how to move, and how to be in my body in the ways. . . that I want to, I want to expand, that I want to move. . . and someone who's going to work with me to co-create that, as opposed to always being the expert and telling me what I need to do in order to accomplish something, because I have. . . experiential knowledge of my body, and I want that to be taken seriously, and not for me to be put in some. . . sort of box.

We imagine a fitness doula as someone who has a capacitating role in a fitness space and who helps support folx in exploring their own bodily potential outside of what is normatively expected. This concept is drawn from the notion of crip doulaship in disability communities, whereby crip doulas mentor other disabled people and facilitate their access to disability community by supporting their navigation in the ableist world (Valentine 2020). Crip doulaship relies on the wealth of knowledge and skills that disabled people possess as a function of having to navigate a world that lacks supportive structures. "Crip wealth" and crip doulaship are both key to access intimacy, in our view, and capture ways in which our bodily freedom can be enhanced in ableist spaces (Valentine 2020, p. 91). Through the lens of access intimacy and crip doulaship, access becomes something we do or practice together, rather than something that we are striving to achieve or complete.

Furthermore, we envision love, care, connection, support, and a move towards pleasure and desire as being part and parcel of our reconceptualisation of fitness. As one participant co-researcher, Skylar, said during a group meeting, "fitness is feeling good in my body" and "when I feel good in my body, that's when I feel like I am fit." This expands what fitness can look/feel like beyond dominant norms and prompts us to reflect on the conditions or elements that can enable people to think about fitness and movement in relation to their own positive bodily experiences and desires. What if people were supported to move in a way that feels good, whatever that means for them, rather than in trying to subscribe to a narrow idea of "exercise"? How can we best support people to move towards the edges of their embodied needs and desires in ways that are capacitating?

## 4. Final Remarks

The fitness world is preoccupied with the ableist, capitalist notion of "moving things forward": harder, better, faster, stronger. The *ReVisioning Fitness* CoP rejects this model in favour of moving together at an artful, variable pace across space, time, and difference. This is one way that access intimacy functioned and continues to function for us. The innovation of this CoP, then, lies not in its speed or forward motion but rather in its ability to meet each of its members where they are (even if that is at home on their couch or bed). Our very process of working on this article operated in crip time, in that it required slowness and care to allow our ideas to develop, urging us to reject the need to always be moving ahead quickly to reach academic deadlines.

Within the delimited container of the project, a CoP that enabled dissent, body-becoming pedagogies, and access intimacies unfolded, a space where team members felt capacitated by the community and the support available therein. We recognise that this work is still in a process of unfolding, shifting, and becoming, and just like all participa-

tory community-based work, we may experience collapse at any point in the journey. At this moment, we are left pondering the "big" questions: How might fitness take lessons from disability culture with its enactment and mobilisation of access intimacy to transform approaches to accessibility? And, building on learnings from *ReVisioning Fitness* and the knowledge held by disabled people, what might the fitness world do to create the conditions for difference-affirming and relational fitness experiences?

We also encourage researchers and leaders in fitness spaces to keep in mind that no experiment with access will materialize in exactly the same way given the fluidity of body-worlds and the fact that the specific bodies and forces acting on them will never be precisely the same, making a prediction of outcomes difficult, if not impossible. While it is feasible to anticipate some access needs (e.g., difference-affirming language use, creating a relaxed space), difference is always making a difference, meaning that when one creates the conditions of possibility for difference to emerge, difference will surface in different ways and in ways specific to individualities and group configurations. Thus, one could replicate or inherit many of the conditions we describe here and have an entirely different outcome, leading us to suggest a general philosophical engagement with the concepts we present rather than strict adherence to our approach.

## 5. Conclusions

Our centring of disability arts and culture helped to create the conditions of possibility for team members to go to the edges of and make meaning from their unique embodied needs, interests, and desires. A difference-affirming context derived from a belief in relationality allowed us to consider the movement practices we wanted to engage in and the conditions under which we might access and experiment with them. Rather than considering fitness as an instrumental praxis for achieving an individualistic or standard ideal of body functionality, our focus on the physical through the lens of disability arts and culture became a creative exploration of perception, movement, and embodiment. This exploration enables us to engage in (re)making the world of fitness through the lens of queering, cripping, and thickening fitness practices. Fitness leaders can learn from disability art making, our relational conceptualisation of CoP, and principles of body-becoming and access intimacy, and we call on them to do so and to open conditions of possibility in fitness through a radical embrace of difference.

**Author Contributions:** Conceptualization, M.B., K.A.B., K.B., C.R., S.P. and T.-L.F.M.; methodology, M.B., K.A.B., C.R. and T.-L.F.M.; formal analysis, M.B. and K.A.B.; investigation, M.B., K.A.B., C.R. and T.-L.F.M.; resources, K.A.B. and C.R.; writing—original draft preparation, M.B., K.A.B. and K.B.; writing—review and editing, M.B., K.A.B., K.B., C.R., S.P. and T.-L.F.M.; supervision, K.A.B., C.R. and T.-L.F.M.; project administration, K.A.B. and M.B.; funding acquisition, K.A.B., C.R. and T.-L.F.M. All authors have read and agreed to the published version of the manuscript.

**Funding:** This research was funded by the Social Sciences and Humanities Research Council of Canada grant number 430-2020-00030.

**Institutional Review Board Statement:** This research was approved by the University of Guelph Research Ethics Board (protocol # 20-10-005, approved 15 December 2020).

**Informed Consent Statement:** Informed consent was obtained from all participants involved in the study.

**Data Availability Statement:** Please contact the corresponding author if you are interested in any of the data mentioned in this article.

**Acknowledgments:** The authors would like to thank the Re•Vision Centre for Art and Social Justice for their support in the multimedia storytelling workshop, specifically the workshop facilitators, Hannah Fowlie and Calla Evans, and the managing director, Ingrid Mündel. The authors would also like to acknowledge the other members of the ReVisioning Fitness team who did not have capacity to contribute to this article but have given in important ways to the project as a whole: these are Bongi Dube, Evadne Kelly, seeley quest, Skylar Sookpaiboon and Paul Tshuma. We are also

grateful for the contributions of Ash McAskill in the early stages of this project as she was integral in connecting the members of this group. Lastly, we thank the guest editors of this special issue, Nadine Changfoot, Carla Rice, and Eliza Chandler, for curating this important collection of articles and reviewing our manuscript.

**Conflicts of Interest:** The authors declare no conflict of interest.

## Notes

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
