# Peer review of "Revisioning Fitness through a Relational Community of Practice: Conditions of Possibility for Access Intimacies and Body-Becoming Pedagogies through Art Making"

_socsci, doi:10.3390/socsci12100584_

Round 1

Reviewer 1 Report

Thank you for the opportunity to review this article. As a queer, trans, and disabled scholar who is invested in "fitness" in various liberatory forms, I appreciate the work the authors do to disrupt norms of exclusion and oppression within these spaces. I particularly enjoyed the four "imagined and theorized alternative offerings to the current fitness climate." Those offer tenets for future researchers and practitioners to explore in the future.

Other praises for the article:

  • I appreciate the attention to citing marginalized scholars and activists throughout the article. Politics of citation are crucial when writing work like this, and I was very pleased to see the diversity of scholars cited here.

  • Overall, the writing is beautiful. I applaud the authors’ ability to coauthor so smoothly, and to articulate complex concepts in ways that are easy to read. This is no small feat.
  • The language used throughout avoids infantilization of disabled, PoC, and/or queer folks, which is huge! Unfortunately, this is not always the norm. I appreciate the authors’ attention to using language that these communities embrace.

Places for language revision:

  • This article includes many terms that I believe require more extensive explanation. Some, like “creative captioning” are likely unfamiliar to general social science audiences. Others, like "thicc" (or “thicc/thick”) read oddly without more context on the racialized nature of the language.

  • “Crip” is a unique political position which I believe requires some background or attention in this article. Not all disability & access folks adopt “crip” language, so a nod to what crip represents (especially in relation to queerness) would be helpful.

Ethical concerns:

  • When describing yourselves as five white cis women, why not engage body size and disability, since these seem crucial to your work? I felt myself immediately asking if any of the academics in this project occupied marginalized embodiments in those categories.

  • Talk about how you recruited the marginalized participant co-researchers. A major issue with participatory research is the potential for exploitation. I’m not saying that you exploited your co-researchers/participants whatsoever. However, more attention to this pitfall and how you avoided it would be helpful. What were the benefits to the multiply marginalized co-researchers that you recruited? Perhaps could be elaborated in relation to the importance of “saying no” as detailed later in the paper.

  • This paper reads as very utopian. As a community-based researcher, I understand the magic that these CoPs can create. However, I also know that they’re messy! I’d like to see you lean into that messiness more directly. What difficult moments have you addressed? In what moments did you experience collapse? Without acknowledging the dark sides of this project, it reads to me as lacking reflexivity—especially considering the identities of the academic scholars compared to the community researchers. You may not have space to include a lot of these narratives, but more attention to the dark sides of this research is key considering the history of researchers exploiting marginalized people.

Points for general revision:

  • As a reader, it would be helpful for you to give a more thorough background (even just a paragraph) about the CoP. You describe the goals very clearly (and in beautiful language), but I could use a couple sentences that describe the “what” of what you actually do. Working out together? Talking in groups about your experiences with fitness? Making art in person? Making art virtually? When you made videos, why did you do it? Some of that background knowledge would really help bring clarity for readers.

  • I recommend hedging the statements at the beginning that frame current fitness communities as oppressive. While many *certainly* are and you are right to critique them, I think it would be helpful to acknowledge that marginalized groups have created many fitness groups/spaces that are explicitly dedicated to evading these issues. As a queer, trans, and disabled person, I have been part of several these groups! Gesturing to a few as models or simply others doing similar work to your CoP would be helpful and acknowledge marginalized labor more fully.

I hope these points of suggested revision are helpful as you move forward with this piece.

Author Response

Please see the attachment for a response to both Reviewer 1 and Reviewer 2. 

Reviewer 2 Report

It is very rare when I feel there are no improvements I can offer a paper I have been invited to review. This is one of those occasions! I commend the authors for producing such a coherent and informative paper about what sounds to be a fascinating project, one I will be sure to follow-up once the anonymity is lifted from this piece. The paper provides an excellent combination of originality, theoretical grounding, and opportunity for readers to learn more about an approach to research that they may not have considered or feel uncertain about how to pursue. I am very much looking forward to seeing this piece published so that I can share it among my networks. Thank you authors!

Author Response

(The authors gave the same response as above.)
